# Rice growth and yield responses to saline water irrigation are related to Na⁺/K⁺ ratio in plants

Priya Lal Chandra Paul[1]*, Afsana Jahan[2], Palash Kumar kundu[1], Debjit Roy[1], Richard W. Bell[3], Md Belal Hossain[1], Rakiba Shultana[4], Mohammad Rezoan Bin Hafiz Pranto[1], Tanjina Islam[2], Sharon E. Benes[5], Md Rafiqul Islam[2]

1 Irrigation and Water Management Division, Bangladesh Rice Research Institute, Gazipur, Bangladesh, 2 Soil Science Division, Bangladesh Rice Research Institute, Gazipur, Bangladesh, 3 Centre for Sustainable Farming Systems, Future Food Institute, Murdoch University, Murdoch, WA, Australia, 4 Agronomy Division, Bangladesh Rice Research Institute, Gazipur, Bangladesh, 5 Department of Plant Science, California State University, Fresno, CA, United States of America

* plcpauliwm@yahoo.com

**Data Availability Statement:** The data underlying the results presented in the study are available as supplemental information.

## Abstract

Rice growth and yield response to salinity can be influenced by the duration and the timing of salt stress. The present study tested the effects of saline water irrigation from vegetative growth to maturity on rice growth and yield and ion concentrations in the straw and root and related them to changes in soil salinity and soil solute potential. The treatments consisted of five levels of saline water irrigation (electrical conductivity ~0.25 (control), 4, 6, 8, and 10 dS m⁻¹) with two rice cultivars (BRRI dhan67 and BRRI dhan99) grown in pots in a rain shelter. Grain weight per pot, dry straw weight, and root weight were significantly reduced with increasing water salinity, but BRRI dhan99 was less affected. With prolonged saline water irrigation, salt concentration increased in the soil and lowered the soil solute potential. Increased saline water induced higher concentrations of Na⁺ in the straw (527–1200 mmol kg⁻¹ at 4–10 dS m⁻¹) relative to the root. By contrast, higher Cl⁻ concentrations accumulated in the root than in the straw. The decrease of K⁺ in the straw and root for increasing salinity was inconsistent, but the Na⁺/K⁺ ratio sharply increased in the straw with higher water salinity. The increased Na⁺/K⁺ explained most grain weight loss due to higher salinity ($R^2 = 0.93$) followed by Na⁺ ($R^2 = 0.87$) and Cl⁻¹ ($R^2 = 0.53$). We conclude that the prolonged saline water irrigation has a cumulative effect on root zone salinity and solute potential that depresses grain yield in rice by increasing the Na⁺/K⁺ ratio in plants.

## Introduction

The spread of salinity in water and soil poses a threat to crop production in many parts of the world. Globally, more than 1000 million hectares of land are affected by different degrees of salinity [1] which leads to loss of agricultural production of around 12 billion US$ annually [2]. It is predicted that around 50% of all arable land will be salinized by 2050 [3] while the

**Funding:** The author(s) received no specific funding for this work.

**Competing interests:** The authors have declared that no competing interests exist.

demand for crop production will be 60–110% higher [4]. Rice is widely grown as a major crop across the globe, as the staple food for nearly 50% of the world's population [5, 6]. The irrigated rice cultivation system requires 40–50% of total water used in agriculture [7]. However, due to the scarcity of fresh irrigation water in many areas irrigation with salty water is being considered for continuing crop production. Large areas of the Ganges Delta remain fallow during the dry season because of high soil salinity and scarcity of freshwater [8]. In the last five years, brackish irrigation water in this area has been used for non-rice crop cultivation. However, the risk of heavy rainfall greatly damages the production of non-rice crops (i.e., sunflower, maize, watermelon, etc.) in this area, while rice cultivation avoids this risk [9]. The river and canal water salinity ($EC_w$) in this area increased from 2–20 dS m$^{-1}$ from January to May [10]. The cultivation of salt-tolerant rice along with the use of saline water could be an alternative to growing rice in this area but the threshold level of salinity that can be tolerated by rice is unknown [9].

Salinity hampers plant growth due to changes in physiological and biological attributes and deterioration of soil physical and chemical properties, resulting in soil degradation and lower crop production [2]. In saline soil, low soil solute potential, ion toxicity, excess availability of ions ($Na^+$, $Cl^-$ and $SO_4^{2-}$) and an imbalance of nutrients are the main causes of poor crop growth and development [11–13]. Effects of increased salt concentration in the root zone include lower photosynthesis, leaf injury, suppressed enzyme activity, and malfunctioning of cell membranes [2]. The negative impact of salinity on rice cultivation has been reported [14–17], however, the adverse effects caused by ionic and osmotic stresses at different stages and for various cultivars of rice have reported contrasting results [18]. However, it is reported that the use of highly-saline water increases sodium ($Na^+$), calcium ($Ca^{2+}$), magnesium ($Mg^{2+}$), sulphates ($SO_4^{2-}$), and chlorides ($Cl^-$) in the soil and plant, which inhibit crop physiological processes and growth, thus decreasing production [19, 20].

Some studies showed that seedling and reproductive growth stages were more sensitive to root zone salinity than tillering [21, 22] while others reported that salinity imposed at the tillering stage reduced shoot dry weight more severely than the salinity at reproductive stages [23]. A study conducted by [24] pointed out that the growth reduction of rice subjected to salinity stress was mainly related to the uptake of excess ions, while [18] reported that the impact of salinity stress on rice growth and yield components was mostly attributed to osmotic stress. Hence, they suggested that further research should resolve the effects of osmotic and ionic stress on rice and apply a range of salinity levels over a longer duration. This gap in understanding prolonged response of salt-tolerant rice varieties to medium and high levels of water salinity in saline soil remains. In addition, it is important to understand the physiological and ionic mechanisms behind the response of different rice cultivars to long-term saline water irrigation. Plant salt tolerance is usually quantified based on the biomass production under short to long-term saline water irrigation. Previous studies reported that increased NaCl concentrations reduced the dry weight of plants, which was minimal in salt-tolerant species and highest in salt-sensitive species [20, 25]. However, salt-tolerant crops have the ability to reduce the $Na^+$ and $Cl^-$ transport to leaves and prevent the build-up of these ions in cytoplasm or cell walls hence minimising salt toxicity in shoots.

A review by Shalhevet [26] on the use of marginal quality water on crop production concluded that the duration of exposure to salinity was more important in determining crop inhibition by salinity than the sensitivities in different growth stages. Zeng, Shannon [27] mentioned that the salt stress at various growth stages can be quantified if the salinization duration is the same.

As the saline water application increases the salt concentration in the soil, yield reduction is generally related to the electrical conductivity of the soil (either EC1:5 or ECe) rather than soil

solute potential [28, 29]. However, recent study conducted by [8, 30, 31] showed that the solute potential of soil solutions better explained effects of salt stress on crop growth and yield than soil electrical conductivity.

The present study hypothesized that the ion concentrations in rice and solute potential in the soil subjected to different levels of water salinity best explained rice response to saline water irrigation. The objectives of our study were to determine the impact of different levels of saline water irrigation on (i) rice growth and yield components, (ii) ion concentration in the shoot and root, and (iii) changes in soil salinity and solute potential.

## Materials and methods

### Experiment description

The pot experiment was conducted in a rain shelter at Bangladesh Rice Research Institute, Gazipur (23°59′25″ N latitude and 90°24′27″ E longitude) from January to May 2022, where the minimum temperature was in February (9.5°C) and the maximum temperature was in April (39.7°C). The pot size was 30 cm in height and 30 cm in diameter containing 10 kg of soil. The pot was filled with soil (0–15 cm layer) collected from a coastal saline area (22.62°N latitude and 89.51°E longitude). The collected soil was air-dried and sieved through a 5 mm diameter mesh. The texture of the soil was silty clay (sand 9%, silt 40%, and clay 51%), with a pH of 7.7, $EC_{1:5}$ of 0.39 dS $m^{-1}$, a nitrogen level of 1.1 g $kg^{-1}$, exchangeable sodium 4.4 cmol $kg^{-1}$, exchangeable potassium 0.9 cmol $kg^{-1}$, and exchangeable calcium 15.9 cmol $kg^{-1}$. The pot was filled with dried soil and puddled with fresh water (0.02 dS $m^{-1}$). Two salt-tolerant rice cultivars, BRRI dhan67, and BRRI dhan99, were used for this experiment. BRRI dhan67 has medium salt-tolerance (tolerates ECe ~ 6–8 dS $m^{-1}$ in the whole life cycle) with -150 days growth duration while BRRI dhan99 was highly salt tolerant (tolerates ECe ~ 8–10 dS $m^{-1}$ in the whole life cycle) but has a longer growth duration (148–157 days) [32]. Thirty-five days old seedlings were transplanted on 2 Feb 2022 in each pot maintaining 4 hills per pot with 2 seedlings per hill. Before transplanting, all fertilizers except urea were mixed with the soil.

After transplanting, tap water (EC~0.25 dS $m^{-1}$) was applied as irrigation up to 25 days when saline irrigation water treatments were imposed. Irrigation water was applied with five different levels of salinity: 0.25 (control), 4, 6, 8, and 10 dS $m^{-1}$. Saline irrigation water was collected in a large jar from a coastal river with an EC of 6 $dSm^{-1}$. Irrigation with saline water was maintained throughout the growing period by adding coastal saline water mixed up with tap water (0.5 $dSm^{-1}$). In each case, sodium chloride was added to get the required salinity level. The pots were organized in a randomized complete block design with three replications. Saline water irrigation was imposed 25 days after transplanting and continued until 15 days before harvesting. BRRI dhan67 and BRRI dhan99 were harvested on 10 May and 20 May 2022, respectively. At harvesting, total grain weight per pot, number of filled and unfilled grains, number of tillers per pot and plant height were recorded.

### Sampling and measurements techniques

From each pot, soil samples were collected at 45 days and at harvesting at 0–15 cm depth to measure $EC_{1:5}$ and the solute potential of soil solutions. Two hills at 45 days and at harvesting were sampled to measure shoot and root dry weight and ion concentration ($Na^+$, $K^+$, and $Cl^-$).

**Measurement of soil water content, $EC_{1:5}$, and solute potential.** The soil water content was determined following the gravimetric method. Initial soil samples were weighed and then samples were oven-dried at 105°C to a constant weight. Soil salinity ($EC_{1:5}$) was measured by a

portable EC meter in a soil water suspension (1:5 soil and water) that had settled for an hour. The following equation was used to estimate the solute potential of the soil solution [8].

$$\Psi s = \frac{-22580 \times EC_{1:5}}{W}$$

Where, $\Psi s$ = solute potential (kPa), $EC_{1:5}$ = electrical conductivity of soil water (dS m$^{-1}$) and W = soil water content (%, w/w).

**Measurement of straw and root Na$^+$, K$^+$ and Cl$^-$ concentrations.** At harvesting, 2 hills (the whole plant) were collected to separate above-ground biomass (straw) and roots to analyze the ion concentrations and ratio of Na$^+$ to K$^+$. Firstly, the collected samples were rinsed in distilled water separately and then dried in an oven at 70˚C for 72 h. The di-acid digestion method was followed to determine the Na$^+$ and K$^+$ concentrations in straw and root [33]. The di-acid contained the mixture of analytical grade concentrated $HNO_3$ and $HClO_4$ acids in a ratio of 5:2. Accordingly, 7 mL of the di-acid was added to 0.5 g of powdered sample in a 50 mL conical flask and soaked overnight inside a fume hood. The following day, the samples were digested in a hot plate (Cole- Parmer, HP 200 Series, Made in U.S.A) at 200˚C for 3 hours until a tiny amount of thick liquid residue was left at the bottom of the conical flask. The samples were then cooled to room temperature, diluted with Ultrapure water (SUEZ, Made in UK) and thoroughly mixed with a vortex mixer. The samples were then filtered using Whatman filter paper 1 and stored in plastic tubes at 4˚C. The concentrations of Na$^+$ and K$^+$ were then measured with a flame photometer (Model: 410, Sherwood). The chloride ion from the plant tissue and soil samples was extracted with $HNO_3$ digestion and water (1:5 soil/water), respectively. About 10 mL analytical grade concentrated $HNO_3$ was added to 0.5 g of dried and powdered rice tissue in a 250 mL conical flask and soaked overnight inside a fume hood for pre-digestion. After pre-digestion, when the sample was almost dissolved, another 10 mL of $HNO_3$ was added. The samples were digested in a block digester (J. P. SELECTA, s. a. u., Made in Spain) at about 100˚C for the first hour and then the temperature was raised to 150–200˚C. The digestion was continued until the content reduced to 2–3 mL and became colorless. After digestion, the samples were cooled to room temperature and the volume was made to 50 ml with ultrapure water (SUEZ, Made in UK). Then, the samples were filtered through Whatman filter paper 1 and stored in plastic tubes at 4 ˚C for further analysis. The chloride concentration in the extractant was determined by titrating with standard $AgNO_3$ using $K_2CrO_4$ as an indicator [34].

## Statistical analysis

All the data generated were analyzed using analysis of variance (ANOVA) and regression analyses for different factors were performed using R software (version 4.3.0). The significant effects of the yield and yield components, agronomic parameters, soil electrical conductivity, soil solute potential, ion concentration, and ion uptake by plants (straw and root) were analyzed using 2-way ANOVA. The comparison of main effects and interaction means were separated by least significant difference (LSD) at $P < 0.05$.

## Results

### Effect of saline water on rice yield and yield components

Total grain weight per pot was decreased by 15, 46, 77, and 87% for irrigation water at 4, 6, 8, and 10 dS m$^{-1}$ relative to the control treatment (Table 1). BRRI dhan99 had a significantly higher grain weight than BRRI dhan67, but there was no interaction between cultivar and salinity response. The number of filled grains decreased with increasing water salinity.

**Table 1. Yield and yield components of BRRI dhan67 and BRRI dhan99 as affected by different level of water salinity.**

| Variety | Salinity level (dS m$^{-1}$) | Total grain weight per pot (g) | Number of filled grain per pot | Number of unfilled grain per pot | Fertility (%) | Effective tiller per pot (no) |
|---|---|---|---|---|---|---|
| BRRI dhan67 | Control | 34.7 | 1654 | 59.0 | 96.5 | 28 |
| | 4 | 29.4 | 1400 | 67.7 | 95.4 | 27 |
| | 6 | 18.5 | 880 | 84.7 | 91.2 | 16 |
| | 8 | 7.6 | 360 | 146.7 | 70.7 | 8 |
| | 10 | 4.1 | 193 | 196.7 | 49.4 | 5 |
| BRRI dhan99 | Control | 35.2 | 1677 | 7.9 | 99.5 | 30 |
| | 4 | 30.0 | 1430 | 22.5 | 98.4 | 28 |
| | 6 | 19.6 | 933 | 39.9 | 95.9 | 17 |
| | 8 | 8.5 | 405 | 45.7 | 89.8 | 8 |
| | 10 | 5.2 | 247 | 60.7 | 80.2 | 7 |
| Treatment means | | | | | | |
| BRRI dhan67 | | 18.8 | 897 | 29.4 | 80.6 | 17 |
| BRRI dhan99 | | 19.7 | 938 | 35.3 | 92.8 | 18 |
| | Control | 35.0 | 1665 | 33.5 | 98.0 | 29.2 |
| | 4 | 29.7 | 1415 | 45.1 | 96.9 | 27.2 |
| | 6 | 19.0 | 907 | 55.5 | 93.5 | 16.8 |
| | 8 | 8.0 | 383 | 89.5 | 80.3 | 8.0 |
| | 10 | 4.6 | 220 | 128.7 | 64.8 | 5.8 |
| ***P* values** | | | | | | |
| Variety | | 0.05 | 0.05 | NS | 0.001 | NS |
| Salinity level | | 0.001 | 0.001 | 0.001 | 0.001 | 0.001 |
| Variety x Salinity level | | NS | NS | NS | 0.001 | NS |
| **LSD$_{0.05}$** | | | | | | |
| Variety | | 0.84 | 40 | - | 2.1 | - |
| Salinity level | | 1.33 | 63.3 | 8.9 | 3.1 | 3.96 |
| Variety × Salinity level | | - | - | - | 4.5 | - |

Compared to the control, the difference was greatest when irrigation water salinity was applied from 6 dS m$^{-1}$ (almost 50% lower) to 10 dS m$^{-1}$. The number of unfilled grains gradually increased with the increase in water salinity. There was no difference in the percentage of fertility between control and salinity at 4 dS m$^{-1}$. The percentage of fertility was much lower with water salinity at 8 and 10 dS m$^{-1}$. BRRI dhan 99 retaiend fertility of panicles better at his salinity levels than BRRI dhan 67. The effective tiller per pot was almost the same in the control and water salinity at 4 dS m$^{-1}$. The irrigation water salinity at 8 and 10 dS m$^{-1}$ decreased the effective tiller per pot (70–80% lower).

## Effect of water salinity on straw and root dry weight

Relative to the control treatment, the water salinity at 4 dS m$^{-1}$ decreased straw dry weight by 15%, whereas the reduction was 45–87% for irrigation water salinity at 6–10 dS m$^{-1}$ (Fig 1A). With the salinity stress, BRRI dhan99 had a higher straw dry weight than BRRI dhan67 (Fig 1B). Root dry weight was significantly lower at water salinity levels 4–10 dS m$^{-1}$ than the control treatment, however, there was a difference in root dry weight between water salinity at 4, 6 or 10 dS m$^{-1}$ (Fig 1C). Root dry weight was also significantly higher with BRRI dhan99 than BRRI dhan67 (Fig 1D).

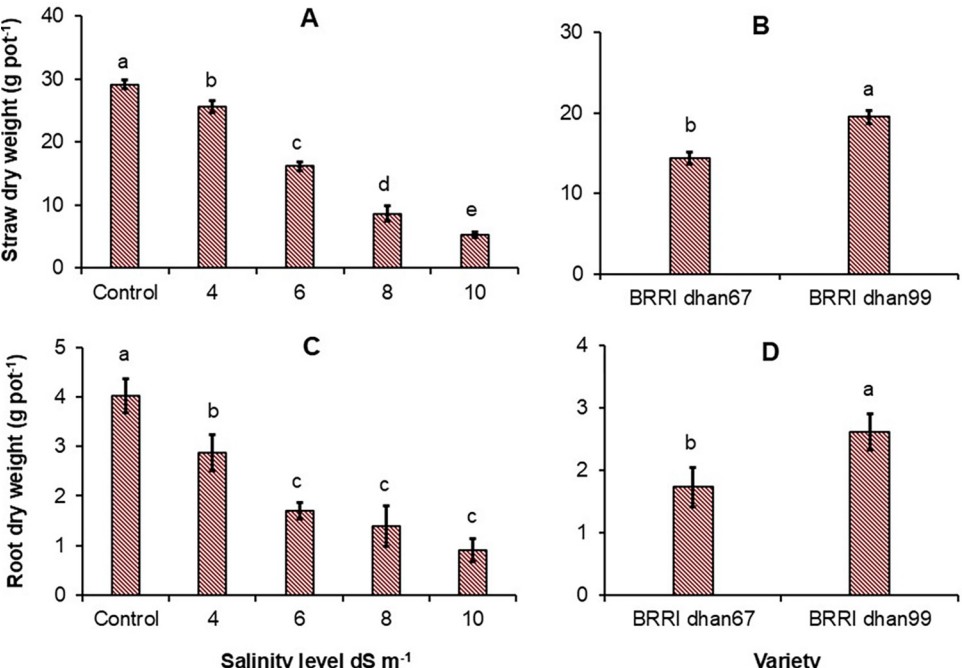

**Fig 1.** Total dry weight of straw (A and B) and root (C and D) as affected by salinity level and variety. The vertical bars in each panel indicate the standard error of the mean. Different lowercase letters above the bars indicate statistically significant differences between treatments at $P < 0.05$.

## Variation of soil electrical conductivity ($EC_{1:5}$) and solute potential (SP) with different water salinity levels

Soil electrical conductivity ($EC_{1:5}$) and solute potential of the soil solution (SP) increased significantly with increasing water salinity at 45 DAT and harvesting (Fig 2A–2D). At 45 DAT and at harvest, among the treatments the soil $EC_{1:5}$ ranged from 0.15 to 0.41 dS m$^{-1}$ (Fig 2A) and 0.18 to 1.3 dS m$^{-1}$ (Fig 2B). At the early stage (45 DAT), soil salinity was minimal, but during the harvest, soil salinity was far greater than water salinity at 6, 8, and 10 dS m$^{-1}$. The solute potential of soil solution decreased (increased negatively) with increasing water salinity throughout the growing season (Fig 2C and 2D). At the beginning, solute potential varied from -150 kPa (control) to -450 kPa (water salinity at 10 dS m$^{-1}$) (Fig 2C). At harvest, the solute potential rapidly decreased to -950 and -1400 kPa for water salinity 8 and 10 dS m$^{-1}$ (Fig 2D) which was significantly lower than control and water salinity at 4 and 6 dS m$^{-1}$.

## Na$^+$ and K$^+$ concentrations and their ratio in straw and root

The dynamics of ion concentrations and their ratio in straw and root are shown in (Fig 3A–3F). Na$^+$ concentration increased in both straw and root with increasing water salinity (Fig 3A and 3B). The highest Na$^+$ concentration was 1188 mmol kg$^{-1}$ at 10 dS m$^{-1}$ followed by 909, 773, and 526 mmol kg$^{-1}$ at water salinity 8, 6, and 4 dS m$^{-1}$. The increasing trend of Na$^+$ in the root was not consistent with increasing water salinity (Fig 3B). There was no difference in Na$^+$ concentration between the control and water salinity at 4 dS m$^{-1}$. The difference in Na$^+$ concentration was much greater between the control (237 mmol kg$^{-1}$) and water salinity at 10 dS m$^{-1}$ (399 mmol kg$^{-1}$). K$^+$ concentration in straw and root was not decreased in all treatments (Fig 3C and 3D). Relative to the control, the K$^+$ concentration was significantly lower when the water salinity was between 8 and 10 dS m$^{-1}$. The ratio of Na$^+$/K$^+$ concentration in straw

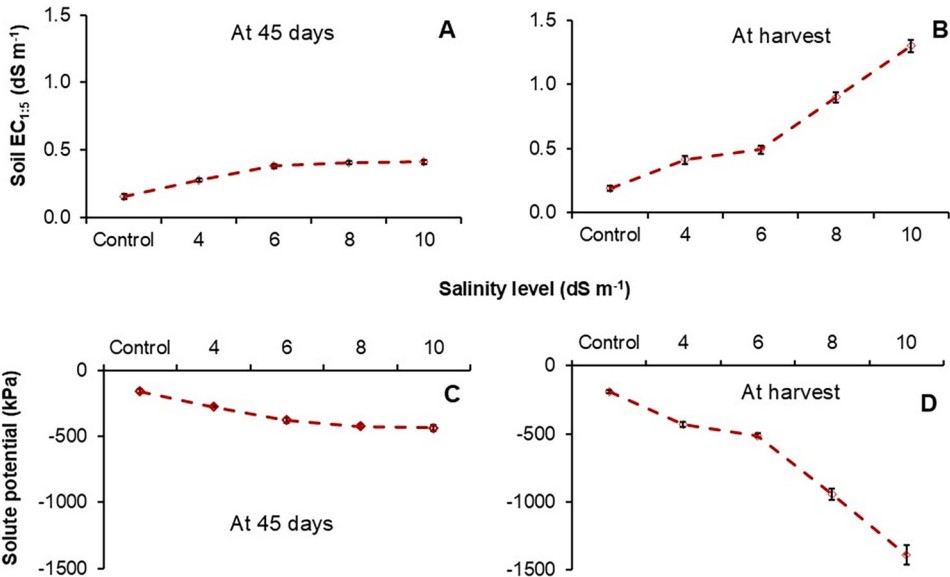

**Fig 2.** Variation in electrical conductivity ($EC_{1:5}$) at 45 days and harvest (A and B) and solute potential of soil solutions at 45 days and harvest (C and D).

gradually increased with increasing water salinity but not in the root (Fig 3E and 3F). The higher water salinity had a higher ratio of $Na^+/K^+$ ion concentration in the straw. However, the ratio of $Na^+/K^+$ concentration in the root was lower than in the shoot.

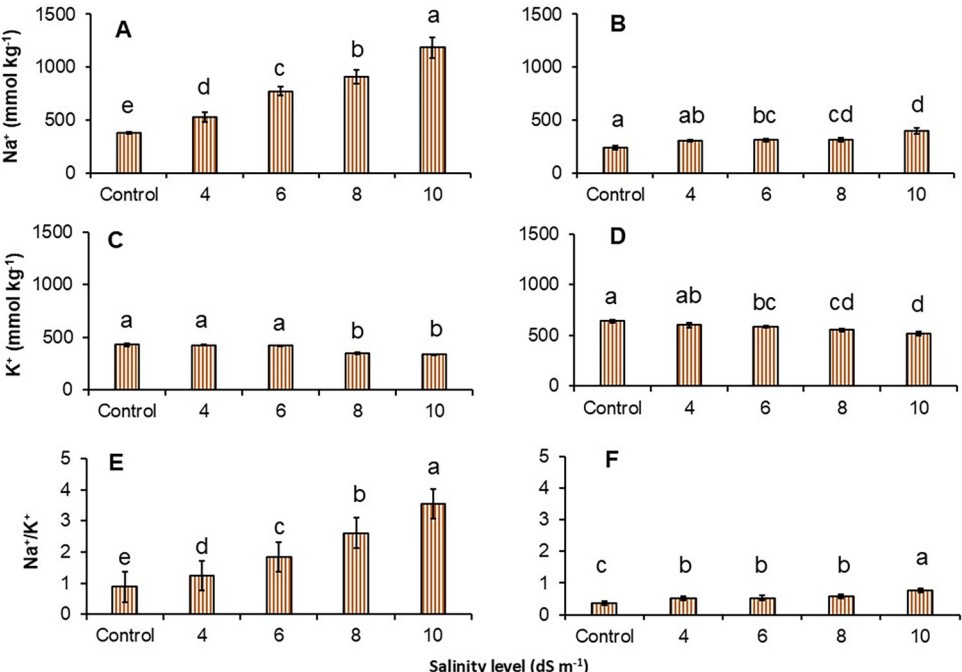

**Fig 3.** Effects of water salinity on the concentration of $Na^+$ in straw (A), root (B), $K^+$ in straw (C), root (D) and $Na^+/K^+$ in straw (E), root (F). Values are means of three replications and each vertical bar indicates the standard error of means. Different lowercase letters above the bars indicate statistically significant differences between treatments at $P < 0.05$.

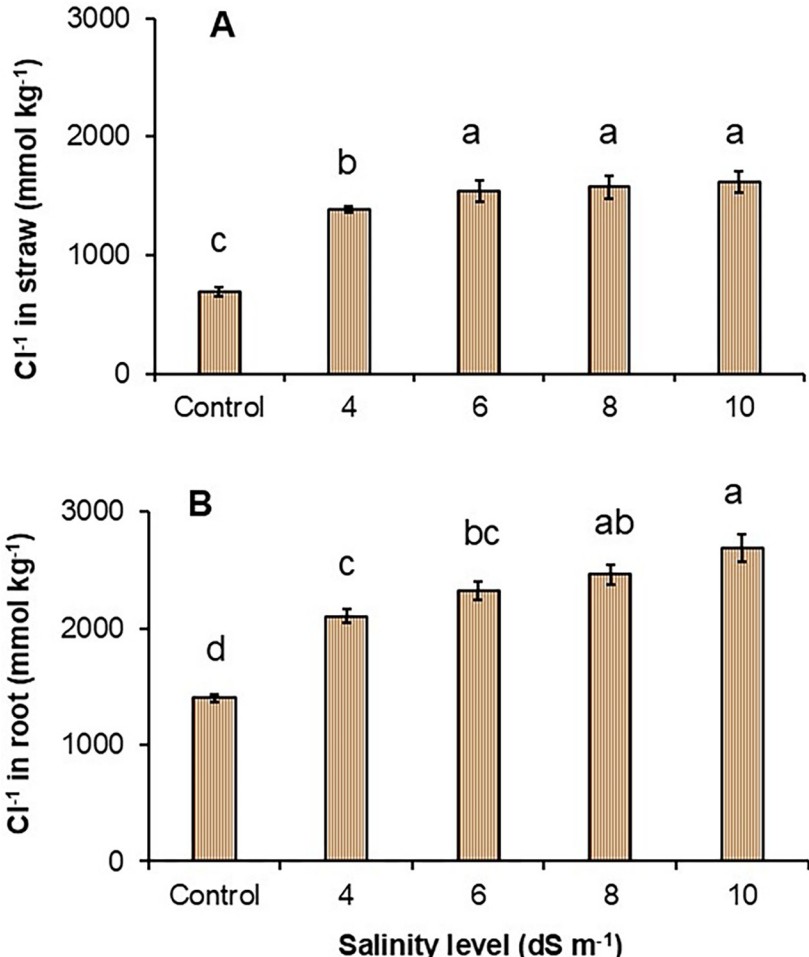

**Fig 4.** Effects of water salinity on the concentration of Cl⁻ in straw (A) and root (B). Values are means of three replications and each vertical bar indicates the standard error of means. Different lowercase letters above the bars indicate statistically significant differences between treatments at $P < 0.05$.

### Cl⁻ concentration in straw and root

The $Cl^{-1}$ concentration increased with increasing water salinity in both straw and root but the Cl⁻ concentration was higher in root than straw (Fig 4A and 4B). Concentrations of Cl⁻ in the straw were about 50–57% higher (1384–1600 mmol kg⁻¹) in the treatments with 4, 6, 8, and 10 dS m⁻¹ than the control treatment (685 mmol kg⁻¹). There was no difference in Cl⁻ concentrations in straw with water salinity of 6, 8, and 10 dS m⁻¹ (Fig 4A). In comparison with the control treatment, the Cl⁻ concentration in the root started to increase when irrigation water salinity was 4 dS m⁻¹. The increment of Cl⁻ concentration in the root was 33, 40, 43, and 48% higher, respectively, in the treatment of water salinity at 4,6,8, and 10 dS m⁻¹ than in the control treatment.

### Relationship between grain weight and ion concentrations

The Na⁺ and Cl⁻ concentrations, and the ratio of Na⁺/K⁺ in the straw were significantly and negatively associated with grain weight, while K⁺ in the straw was positively correlated with the grain weight (Fig 5A–5D). The ratio of Na⁺/K⁺ concentration in the straw explained the

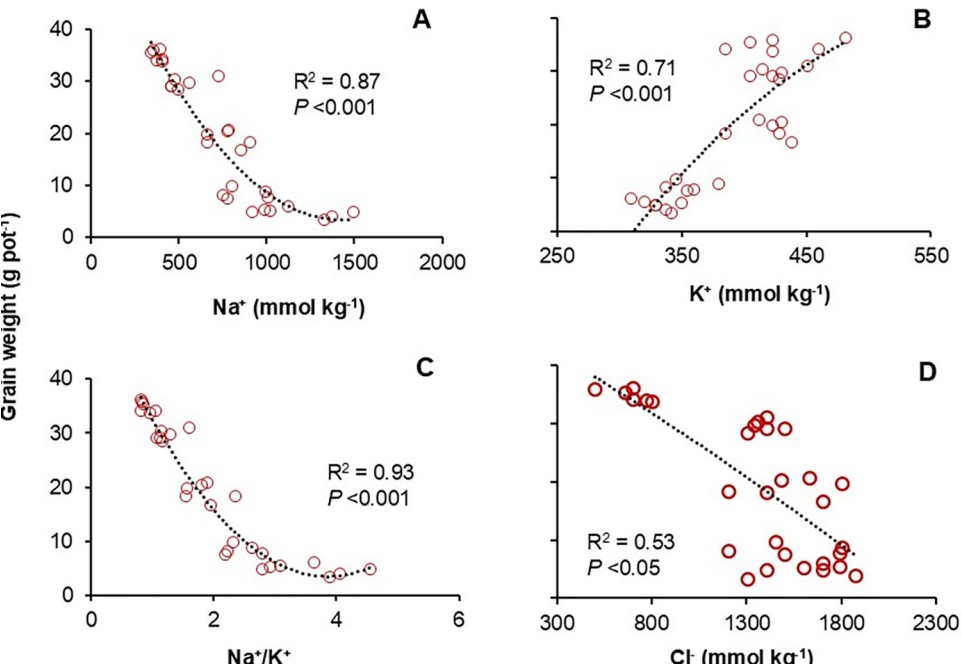

**Fig 5.** Correlation of Na$^+$ (A), K$^+$ (B), Na$^+$/K$^+$ (C) and Cl$^{-1}$ (D) in straw with grain weight in 2022. Each plot represents 30 data points.

highest proportion of variation in grain weight, about 93% (Fig 5C), while Na$^+$ concentration in the straw explained 87% of variation in grain weight (Fig 5A). K$^+$ concertation in the straw explained 71% of the variation in grain weight (Fig 5B). The Cl$^-$explained the least (53%) variation of grain weight (Fig 5D).

## Discussion

Saline water irrigation (brackish) has been used for various crops in many areas of the world [26]. However, using brackish water has potential undesirable consequences, including crop failure. In the present study, crop responses to prolonged saline water irrigation (from vegetative to grain filling) were lower yield, reduced shoot and root weight, higher soil salinity, lower soil solute potential, and increased ion concentrations in the shoot and root. Generally, rice is considered sensitive to moderate salt tolerance depending on cultivars and study conditions [31, 35, 36] and rice yield can be decreased by about 12% by increasing each unit of salinity (dS m$^{-1}$) above the root zone salinity of 3 dS m$^{-1}$ [37]. In this study, compared to freshwater salinity, minimal grain yield loss was caused with the water salinity at 4 dS m$^{-1}$ but there was a more detrimental effect on yield of both varieties when increasing the salinity level (Table 1). The reduction of grain weight arose mostly from the decrease in the number of tillers per pot and the number of filled grains per panicle. Previous studies also reported that higher levels of saline water irrigation cause significant diminishing of growth parameters and grain weight [5, 38, 39]. These irrigation water salinity effects on plant growth and grain yield were related to increasing soil salinity, lowering the soil solute potential [30], and increasing Na$^+$ uptake by roots and shoots [18].

Salinity imposed for a long time has had a significant influence on rice shoot and root growth. Strong differences were observed in rice straw dry weight (at harvest) with salinity at 4–10 dS m$^{-1}$ compared to no stress. One of the explanations for declining straw weight with

higher salinity was related to inhibited plant height and number of tillers, which results in an overall decrease in above-ground straw weight [40, 41]. The lower solute potential (more negative) surrounding the root zone (Fig 2D) could limit the uptake of water by plants resulting in poor crop growth and development [8]. In our study, root dry was similar with water salinity at 6–10 dS m$^{-1}$ (Fig 1) which indicates that salt resistance genotypes can have mechanisms to adjust osmotic stress and the extrusion of Na$^+$ ions [42]. Between the two varieties, BRRI dhan99 showed more resistance to salt stress with a higher straw and root dry weight than BRRI dhan67. This can be explained that salt resistance genotypes have evolved mechanisms to limit the uptake and translocation of ions in the shoots and roots as an adaptation to salinity [43, 44]. However, the greater tolerance of BRRI dhan99 was not reflected in greater grain yield.

One of the main concerns for using saline water is salt accumulation in the root zone of soil which ultimately decreases the solute potential and hence interacts closely with crop response. This increased soil salinity was attributed to the lower grain yield. Rather than increasing soil salinity, osmotic stress (lower solute potential) usually occurs with saline water irrigation [11, 45] which inhibits water uptake by plants. In this study, solute potential gradually increased with increasing salinity level, and the solute potential was around -1500 kPa (close to the wilting point) when water salinity was 10 dS m$^{-1}$. The adverse effects of lower solute potential were credited to reduced plant growth and yield [10, 18].

With saline water irrigation, rice shoots absorb Na$^+$ from the soil solution, resulting in plants being exposed to ionic stresses. Increasing the salt stress indicated a higher amount of Na$^+$ concentration in the rice straw (Fig 3A). Plant growth responses to salt stress for a long period involve prolonged transpiration which causes salt build-up in the leaves [16]. Therefore, a specific ion toxicity occurs in the leaves which leads to a reduction of leaf expansion, growing cells, and biomass accumulation. Matoh, Kairusmee [24] reported that rice growth and development reductions exposed to NaCl were mainly because of excess ions, which supports the present study. On the other hand, Na$^+$ accumulation by roots was inconsistent, which suggests that irrespective of increased salt stress, there was little increased Na$^+$ in the roots. This can be happened because root growth is less sensitive under saline conditions than shoot growth [46, 47]. Also, roots are directly in contact with soil solutions and start encountering mechanisms to protect root growth reduction through water uptake and available materials. Higher Na$^+$ concentration in the shoot reduced the K$^+$ accumulation in both shoot and root (Fig 3C and 3D) which has been evidenced in the previous study because of the antagonistic effect of high salinity stress [44]. It can be further clarified that higher levels of salinity lead to lower K$^+$ mainly caused by membrane depolarisation and displacement by Na$^+$ [48]. In the present study, it appeared higher K$^+$ in the root than in the straw, which was contrary to an earlier study conducted by [49]. Chloride (Cl$^-$) is an important element for plant growth up to a certain level (ranging between 350 and 1000 mg kg$^{-1}$). However, the excessive amount of Cl$^-$ due to saline water irrigation or saline soils can be toxic to plants and severely deteriorate plant function and yield [50]. In the present study, high saline water irrigation from 4–10 dS m$^{-1}$ had significantly higher Cl$^-$ content compared to the control but saline water over 4 dS m$^{-1}$ did not change the Cl$^-$ level in the straw. A similar trend was observed in the roots, but the accumulation of Cl$^-$ was much higher in the root systems than in straw (Fig 4A and 4B). Crop response to the higher Cl$^-$ concentration causes lower nitrate uptake and leaf burn [51].

The changes in Na$^+$ and K$^+$ subsequent to increasing the ratio of Na$^+$ and K$^+$ in the straw and root (Fig 3E and 3F). The Na$^+$/K$^+$ ratio is a critical indicator of a plant's ability to maintain ionic balance and an important consideration in understanding the relative toxicities under saline conditions. The higher Na$^+$/K$^+$ ratio with increasing water salinity could be attributed to nutritional imbalance, metabolic disorder, and inefficient stomatal regulation, resulting in

poor plant growth and yield [44, 52]. Lots of studies have been done on the effect of salt stress on plant growth and yield. However, there was a contrasting result from these studies. For example, one report pointed out that the growth and yield reduction of rice were mainly related to excess ion concentration [24] but another report highlighted that salinity-induced rice growth due to osmotic stress [53]. There was a report on $K^+$ increasing or decreasing due to salt stress. Matoh, Kairusmee [24] mentioned that there was a higher $K^+$ in some Indica-type rice roots, whereas it was lower in Japonica rice. In the present study, the reduction of rice grain weight was mainly due to increased $Na^+/K^+$ ($R^2 = 0.93$), $Na^+$ ($R^2 = 0.87$), and $Cl^{-1}$ concentration ($R^2 = 0.53$) (Fig 5A, 5C and 5D).

Across the globe, huge areas of land, especially in the coastal area remain fallow due to high soil and irrigation water salinity. The coastal tidal surge, sea water intrusion and saline water irrigation during the dry season accumulate substantial salts ($Na^+$, $Ca^{++}$, and $Mg^{++}$) in the soil which negatively associates with poor crop growth and yield. Several studies have been done to develop physiological mechanism of salt tolerance cultivars to address the detrimental effects of salinity [2, 53–56]. To increase the rice production in the salt-affected area, Bangladesh Rice Research Institute (BRRI) has developed salt-tolerant rice varieties BRRI dhan67 and BRRI dhan99 where $Na^+$ exclusion mechanism was responsible for both cultivars [43, 57]. However, BRRI dhan99 shows a higher level of tolerance under salt stress conditions because of its compartmentalization mechanism and anthocyanin coloration characteristics which attributed to higher yield than BRRI dhan67 in the regional yield trial [57]. In our study, BRRI dhan99 showed significantly higher yield and lower ion accumulation in saline conditions compared to BRRI dhan67. Recent studies in the Ganges Delta suggested to store more fresh-water in the canal and river and grow salt-tolerant cultivars during the dry season to minimize climate risk and increase cropping intensity [58–61]. Therefore, salt-tolerant BRRI dhan99 can be grown in the saline prone area for sustainable rice production and ensuring food security.

## Conclusions

Increasing irrigation water salinity above 4 dS m$^{-1}$ drastically reduced the rice grain weight, tiller numbers, filled grain, straw, and root dry weight per pot. Saline water irrigation also increased soil salinity and lowered soil solute potential. The ion concentrations ($Na^+$ and $Cl^-$) increased in straw and root with increasing water salinity, irrespective of rice cultivars, while $K^+$ was slightly decreased. The ratio of $Na^+$ and $K^+$ sharply increased in the shoot due to increased salinity. The reduction of rice grain yield due to water salinity was highly correlated with increased ionic components ($Na^+$ and $Cl^-$) of the salt stress. The findings of the present study contribute to predict grain yield in rice under saline conditions with varying degrees of accuracy and their implications for breeding programs focused on salinity tolerance.

## Supporting information

**S1 Data.**
(DOCX)

## Acknowledgments

We would like to give thanks to Soil Science and Agronomy Division laboratory and Irrigation and Water Management Division, Bangladesh Rice Research Institute, Bangladesh for their facility to analyse soil and plant samples. We are grateful to Md. Abdullah Al Mamun, Senior Scientific Officer, Statistics Division, Bangladesh Rice Research Institute for his suggestions on doing statistical analysis and Dr. Mohammad Akhlasur Rahman, Principal Scientific Officer,

Plant Breeding Division, Bangladesh Rice Research Institute for his valuable advice to prepare the manuscript.

## Author Contributions

**Conceptualization:** Priya Lal Chandra Paul.

**Data curation:** Priya Lal Chandra Paul, Afsana Jahan, Palash Kumar kundu, Rakiba Shultana, Mohammad Rezoan Bin Hafiz Pranto, Tanjina Islam.

**Formal analysis:** Priya Lal Chandra Paul, Afsana Jahan, Rakiba Shultana, Tanjina Islam.

**Investigation:** Afsana Jahan.

**Methodology:** Priya Lal Chandra Paul, Palash Kumar kundu, Richard W. Bell.

**Supervision:** Priya Lal Chandra Paul, Afsana Jahan, Palash Kumar kundu, Mohammad Rezoan Bin Hafiz Pranto.

**Writing – original draft:** Priya Lal Chandra Paul.

**Writing – review & editing:** Debjit Roy, Richard W. Bell, Md Belal Hossain, Rakiba Shultana, Sharon E. Benes, Md Rafiqul Islam.

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
