## [Decision Letter · Decision Letter 0]

13 Aug 2024

PONE-D-24-29091Rice growth and yield responses to saline water irrigation are caused by increased ion concentration in straw and rootsPLOS ONE

Dear Dr. Paul,

Thank you for submitting your manuscript to PLOS ONE. After careful consideration, we feel that it has merit but does not fully meet PLOS ONE’s publication criteria as it currently stands. Therefore, we invite you to submit a revised version of the manuscript that addresses the points raised during the review process.

We look forward to receiving your revised manuscript.

Kind regards,

Yousef Alhaj Hamoud, Ph.D.

Academic Editor

PLOS ONE

Journal Requirements:

We would like to give thanks to Soil Science and Agronomy Division laboratory and Irrigation and Water Management Division, Bangladesh Rice Research Institute, Bangladesh for their facility to analyse soil and plant samples. We are grateful to the Ministry of Agriculture, Government of Bangladesh for funding support. 

4. In the online submission form, you indicated that Data will be available on request.

5. Please include your tables as part of your main manuscript and remove the individual files. Please note that supplementary tables (should remain/ be uploaded) as separate "supporting information" files

Reviewers' comments:

Reviewer's Responses to Questions

**Comments to the Author**

1. Is the manuscript technically sound, and do the data support the conclusions?

Reviewer #1: Yes

Reviewer #2: Yes

Reviewer #3: No

2. Has the statistical analysis been performed appropriately and rigorously? 

Reviewer #1: Yes

Reviewer #2: Yes

Reviewer #3: Yes

3. Have the authors made all data underlying the findings in their manuscript fully available?

Reviewer #1: Yes

Reviewer #2: Yes

Reviewer #3: Yes

4. Is the manuscript presented in an intelligible fashion and written in standard English?

Reviewer #1: Yes

Reviewer #2: Yes

Reviewer #3: Yes

5. Review Comments to the Author

**Reviewer #1:** Dear Editor,

Thank you for selecting me as a potential reviewer for this manuscript. I appreciate the opportunity to review this work, which presents a significant and novel investigation into the effects of saline water irrigation on rice growth and yield. The study's focus on ion accumulation in the straw and roots and its relation to changes in soil salinity and soil solute potential is both timely and relevant, particularly in the context of increasing salinity stress in rice cultivation areas. This manuscript explores the impact of saline water irrigation on rice growth, yield, and ion concentration in straw and roots. The study uniquely examines how different levels of salinity affect two rice cultivars, BRRI dhan67 and BRRI dhan99, and finds that the latter is more tolerant to salinity stress. The research demonstrates that increased salinity leads to a significant accumulation of sodium in the straw and chloride in the roots, while also reducing soil solute potential. The study’s analysis reveals that the Na+/K+ ratio in the straw is a critical determinant of grain weight, highlighting the importance of ion balance in mitigating the adverse effects of salinity.

While the manuscript presents valuable findings, I would like to offer a few suggestions that could further enhance the quality of this work:

The study effectively measures ion concentrations in both straw and roots across different salinity levels. However, the manuscript would benefit from a more detailed explanation of how the ion concentrations were measured, particularly the techniques and instruments used. This would provide greater clarity and ensure the reproducibility of the results.

The research investigates the effects of saline water irrigation on two rice cultivars, yet the manuscript does not discuss the genetic or physiological basis for the observed differences between BRRI dhan67 and BRRI dhan99. A discussion on potential genetic markers or physiological traits that contribute to the salt tolerance in BRRI dhan99 would significantly enrich the findings and provide insights for future breeding programs.

The manuscript discusses the reduction in grain weight, straw weight, and root weight under saline water irrigation. However, the study lacks information on the statistical power of the experiments, including the sample size and the replication of treatments. It is essential to provide details on these aspects to ensure that the results are statistically robust and can be generalized beyond the specific conditions of the study.

The study mentions the impact of saline water on soil solute potential but does not adequately address the spatial variability of soil properties within the pots. Salinity can vary significantly within the root zone, affecting plant responses. A more rigorous approach, such as using soil moisture sensors or conducting soil sampling at different depths, would strengthen the conclusions related to soil solute potential and its impact on rice growth.

The Introduction section contains several old and traditional references. To enhance the quality and relevance of the paper, I recommend updating these references with more recent and relevant citations. This will not only strengthen the background information but also align the study with the latest research developments in the field. Below are some suggested references that could be considered for inclusion:

Mitigation of salinity stress in barley genotypes with variable salt tolerance by application of zinc oxide nanoparticles

An overview of hazardous impacts of soil salinity in crops, tolerance mechanisms, and amelioration through selenium supplementation

Foliar application of ascorbic acid enhances salinity stress tolerance in barley (Hordeum vulgare L.) through modulation of morpho-physio-biochemical …

Melatonin-Induced Salinity Tolerance by Ameliorating Osmotic and Oxidative Stress in the Seedlings of Two Tomato (Solanum lycopersicum L.) Cultivars

Ameliorative Effects of Exogenous Potassium Nitrate on Antioxidant Defense System and Mineral Nutrient Uptake in Radish (Raphanus sativus L.) under Salinity …

Anatomical adaptations and ionic homeostasis in aquatic halophyte Cyperus laevigatus L. under high salinities

Structural and Functional Determinants of Physiological Pliability in Kyllinga brevifolia Rottb. for Survival in Hyper-Saline Saltmarshes

… mycoides PM35 reinforces photosynthetic efficiency, antioxidant defense, expression of stress-responsive genes, and ameliorates the effects of salinity stress in maize

**Reviewer #2:** This manuscript is well-structured, the methods are sound, the results are comprehensive, and the discussion provides a thoughtful interpretation of the findings. The study contributes to the understanding of the physiological and ionic responses of rice to long-term saline water irrigation, which is relevant for addressing the challenges of salinity in rice production. However, I suggest minor revisions that would help to improve the clarity, flow, and overall presentation of the manuscript.

Title:

The current title "Rice growth and yield responses to saline water irrigation are caused by increased ion concentration in straw and roots" is a bit lengthy and technical. Consider a more concise and clear title, such as:

Introduction:

- Consider adding a sentence or two to provide more context on the use of saline water for rice cultivation in specific regions, such as the Ganges Delta, where this study was conducted.

- You could also briefly mention the importance of understanding the physiological and ionic mechanisms behind the response of different rice cultivars to long-term saline water irrigation.

Materials and Methods:

- in the sentence “……….were analyzed using 2-way ANOVAs” ANOVAs should be ANOVA, as it is one analysis or one design not two ANOVA designs.

- “………were separated by least significant difference (LSD) at P = 0.05 ” ) … P = 0.05 should be

at P < 0.05.

Results:

-For the figures depicting mean comparisons, such as Figures 1, 3, and 4, I would recommend adding the following statement in the figure captions:

"Different lowercase letters above the bars indicate statistically significant differences between treatments at P < 0.05."

- When presenting the data on ion concentrations in the plant tissues, you could include a brief comparison of the patterns observed between the two rice cultivars.

Discussion:

- In the first paragraph, you could start by briefly summarizing the key findings of the study before delving into the discussion.

- Consider adding a paragraph or two to discuss the potential practical implications of your findings for the management of saline water irrigation in rice cultivation.

References:

- The reference list appears comprehensive and up-to-date, covering the relevant literature on the topic. However, you may want to double-check the formatting of the references to ensure they adhere to the PLOS ONE style guide.

- Additionally, consider adding a few more recent references (published within the last 2-3 years) to further strengthen the context and relevance of your study within the current state of knowledge in this field.

**Reviewer #3:** This MS has very limited novelty and its very basic experiment. A lot of published work is available online. I have some questions about this:

1. How do different levels of saline water irrigation (EC 4, 6, 8, and 10 dS m-1) affect the growth parameters such as grain weight, dry straw weight, and root weight in two rice cultivars (BRRI dhan67 and BRRI dhan99)?

2. What makes BRRI dhan99 less affected by increasing water salinity compared to BRRI dhan67? Are there specific physiological or genetic traits that contribute to its tolerance?

3. How does prolonged saline water irrigation influence ion concentrations (Na+, Cl-, K+) in the straw and root of rice plants, and what are the implications for soil salinity and soil solute potential?

4. What is the relationship between the Na+/K+ ratio in the straw and the grain weight of rice? How does this ratio serve as a potential indicator of rice yield under saline conditions?

5. Why does Na+ accumulate more in the straw than in the root, while Cl- accumulates more in the root than in the straw? What mechanisms are responsible for this differential ion distribution?

6. How does the accumulation of salts in the soil from saline water irrigation affect the soil's physical and chemical properties, particularly soil salinity and solute potential?

7. How accurately can the Na+/K+ ratio, Na+ concentration, and Cl- concentration be used to predict grain yield in rice under saline conditions? What are the implications for breeding programs focused on salinity tolerance?

8. What are the critical salinity thresholds (dS m-1) beyond which rice growth and yield are significantly compromised, and how do these thresholds vary between the two rice cultivars studied?

9. What are the long-term effects of saline water irrigation on rice yield and soil health if such practices are continued across multiple growing seasons?

10. How can the findings of this study be applied to improve rice cultivation practices in regions with high salinity, and what management strategies can be recommended to mitigate the negative effects of salinity on rice production.

6. PLOS authors have the option to publish the peer review history of their article (what does this mean?). If published, this will include your full peer review and any attached files.

Reviewer #1: **Yes: **Muhammad Hamzah Saleem

Reviewer #2: **Yes: **Ahmed M. Abdelghany

Reviewer #3: No

---

## [Author Response · Author response to Decision Letter 0]

7 Sep 2024

We have addressed all points raised by the academic editor and reviewer(s).

---

## [Decision Letter · Decision Letter 1]

7 Oct 2024

Rice growth and yield responses to saline water irrigation are related to Na+/K+ ratio in plants

PONE-D-24-29091R1

Dear Dr. Priya Lal Chandra Paul,

We’re pleased to inform you that your manuscript has been judged scientifically suitable for publication and will be formally accepted for publication once it meets all outstanding technical requirements.

Kind regards,

Yousef Alhaj Hamoud, Ph.D.

Academic Editor

PLOS ONE

Additional Editor Comments (optional):

Reviewers' comments:

Reviewer's Responses to Questions

**Comments to the Author**

1. If the authors have adequately addressed your comments raised in a previous round of review and you feel that this manuscript is now acceptable for publication, you may indicate that here to bypass the “Comments to the Author” section, enter your conflict of interest statement in the “Confidential to Editor” section, and submit your "Accept" recommendation.

Reviewer #2: All comments have been addressed

Reviewer #3: All comments have been addressed

2. Is the manuscript technically sound, and do the data support the conclusions?

Reviewer #2: Yes

Reviewer #3: Yes

3. Has the statistical analysis been performed appropriately and rigorously? 

Reviewer #2: Yes

Reviewer #3: Yes

4. Have the authors made all data underlying the findings in their manuscript fully available?

Reviewer #2: Yes

Reviewer #3: Yes

5. Is the manuscript presented in an intelligible fashion and written in standard English?

Reviewer #2: Yes

Reviewer #3: Yes

6. Review Comments to the Author

Reviewer #2: All comments were addressed. I think the paper can be accepted in the current version.

They authors have done good work towards the addressing for the questions and comments raised.

Reviewer #3: All coments have been addressed. Congrates in advance for your publication in PLOS ONE and appreciate your effort in improve the MS.

7. PLOS authors have the option to publish the peer review history of their article (what does this mean?). If published, this will include your full peer review and any attached files.

Reviewer #2: **Yes: **Ahmed Abdelghany

Reviewer #3: **Yes: **Muhammad Usman

---

## [Editor Report · Acceptance letter]

11 Oct 2024

PONE-D-24-29091R1 

PLOS ONE

Dear Dr. Paul, 

I'm pleased to inform you that your manuscript has been deemed suitable for publication in PLOS ONE. Congratulations! Your manuscript is now being handed over to our production team.

Kind regards, 

on behalf of

Dr. Yousef Alhaj Hamoud 

Academic Editor

PLOS ONE